# Eczema Care Online: development and qualitative optimisation of an online behavioural intervention to support self-management in young people with eczema

Kate Greenwell ,[1] Daniela Ghio ,[2] Katy Sivyer ,[1,3] Mary Steele,[1] Emma Teasdale ,[4] Matthew J Ridd ,[5] Amanda Roberts,[6] Joanne R Chalmers ,[6] Sandra Lawton,[7] Sinead Langan ,[8] Fiona Cowdell ,[9] Emma Le Roux,[10] Sylvia Wilczynska,[4] Hannah Jones,[11] Emilia Whittaker,[11] HC Williams,[6] Kim Suzanne Thomas ,[6] Lucy Yardley,[1,12] Miriam Santer,[4] Ingrid Muller [4]

**Correspondence to**
Dr Kate Greenwell;
K.Greenwell@soton.ac.uk

## ABSTRACT

**Objectives** To describe the development of Eczema Care Online (ECO), an online behaviour change intervention for young people with eczema (phase I); and explore and optimise the acceptability of ECO among this target group using think-aloud interviews (phase II).

**Methods** Theory-based, evidence-based and person-based approaches to intervention development were used. In phase I, a qualitative systematic review and qualitative interviews developed an in-depth understanding of the needs and challenges of young people with eczema. Guiding principles highlighted key intervention design objectives and features to address the needs of this target group to maximise user engagement. Behavioural analysis and logic modelling developed ECO's hypothesised programme theory. In phase II, qualitative think-aloud interviews were carried out with 28 young people with eczema and the intervention was optimised based on their feedback.

**Results** The final intervention aimed to reduce eczema severity by supporting treatment use (emollients, topical corticosteroids/topical calcineurin inhibitors), management of irritants/triggers, emotional management and reducing scratching. Generally, young people expressed positive views of intervention content and design in think-aloud interviews. Quotes and stories from other young people with eczema and ECO's focus on living with eczema (not just topical treatments) were valuable for normalising eczema. Young people believed ECO addressed knowledge gaps they had from childhood and the safety information about topical corticosteroids was reassuring. Negative feedback was used to modify ECO.

**Conclusions** A prototype of the ECO intervention was developed using rigorous and complementary intervention development approaches. Subsequent think-aloud interviews helped optimise the intervention, demonstrated ECO is likely to be acceptable to this target group, and provided support for our guiding principles including key design objectives and features to consider when

## Strengths and limitations of this study

► Our rigorous development using complementary theory-based, evidence-based and person-based approaches to intervention development helped ensure the intervention was acceptable and engaging to our sample of young people with eczema.

► Our multidisciplinary intervention development group, including patient and public involvement, ensured that the content was evidence based, that advice was feasible, and that the perspectives of people living with eczema were considered throughout the whole development process.

► Although we were able to recruit a good range of ages, genders and eczema severities for the think-aloud interview study, we did not collect information on participant ethnicity, socioeconomic status, or educational level, which is important for exploring whether the intervention is engaging for all participant groups.

► As think-aloud interviews explore participants' immediate reactions to the intervention, they cannot tell us how people would use the intervention over time or explore engagement with the wider behavioural goals.

developing interventions for this population. A randomised controlled trial and process evaluation of the intervention is underway to assess effectiveness and explore user engagement with the intervention's behavioural goals.

## INTRODUCTION

Atopic eczema (also known as atopic dermatitis) is the most common type of dermatitis/eczematous inflammation and will be referred to from here on as just 'eczema' in

accordance with the nomenclature of the World Allergy Organisation.[1] Eczema is a common skin condition that usually begins in childhood,[2] but for many, the symptoms (dry, sore, itchy skin) can persist into adolescence and adulthood.[3] Eczema management focuses on identification and avoidance of irritants/triggers that may exacerbate eczema symptoms, regular use of emollients to restore skin barrier function and topical corticosteroids or topical calcineurin inhibitors (TCIs) to treat flare-ups.[4]

Eczema management can be particularly challenging in adolescence and early adulthood. Young people report a lack of knowledge regarding their eczema and treatments, and advice provided can often conflict with their own eczema experiences and a competing desire to maintain a 'normal' adolescent life.[5–9] These factors may explain why adherence to topical treatments presents challenges for this age group and many report not using their treatments as prescribed.[7]

Systematic reviews of self-management interventions for people with eczema[10 11] identified that although a few studies evaluated interventions have been developed for parents/carers of children with eczema, only two studies evaluated interventions for children and adolescents.[12 13] Both interventions were delivered face to face and only one reported tailoring their intervention to this age group.[12] To address this research gap and the need to improve self-management support for people living with eczema, the Eczema Care Online (ECO) programme aimed to develop two online behavioural interventions: one for young people with eczema (13–25 years) and one for parents and carers of children (0–12 years) with eczema (www.nottingham.ac.uk/eco).[14 15]

First, we aimed to describe the development of the ECO intervention for young people with eczema (phase I). Second, we aimed to explore and optimise the acceptability of the ECO intervention among young people with eczema (phase II). This article highlights key psychosocial needs of young people with eczema and intervention features to consider when developing behavioural interventions for this group.

## INTERVENTION DEVELOPMENT METHODOLOGY

We used theory-based, evidence-based and person-based approaches to develop the online intervention.[16–18] Guidelines for developing complex interventions emphasise that interventions should be informed by reviews of the current evidence-base, appropriate theory, and an in-depth understanding of the context in which the intervention will be implemented.[19] The person-based approach to intervention development uses iterative qualitative research to understand and accommodate the perspectives of the intervention's target group.[20]

Intervention development was carried out in two phases. In phase I, we collated and synthesised evidence relating to patient behaviours that are likely to reduce eczema severity and the perspectives of young people with eczema (person-based and evidence-based approach).

This evidence guided decisions regarding the intervention's target behaviours and provided us with an in-depth understanding of the key issues, needs and behavioural challenges of this target group. Theory-based approaches (behavioural analysis and logic modelling) were used to develop and illustrate the intervention's hypothesised programme theory, that is, the hypothesised mechanisms of action by which the intervention components exert their effects.[19 21] In phase II, we carried out iterative qualitative think-aloud interviews to gather user feedback on the intervention prototype and optimise it based on this feedback.[22] The methods and results for each phase are reported below.

Both phases were guided by a multidisciplinary intervention development group, which comprised 18 members including patient and public involvement (PPI), dermatologists, a nurse consultant, researchers with an interest in eczema, General Practitioners, health psychologists, and experts in intervention development, writing patient-friendly health information and long-term conditions in adolescents. Through regular meetings and reviewing documents, this group guided the design of the research, helped with the interpretation of the research findings and provided detailed feedback on the intervention plans, written content, website design and prototypes for both online interventions (young people and parents and carers of children with eczema). The intervention development process is illustrated in figure 1.

### Patient and public involvement

Two mothers of children and young people with eczema (one of whom had eczema herself and helps run an eczema support group (AR)) were part of our multidisciplinary intervention development group. Both were coapplicants on the research grant application, helping to identify the research topic and develop research questions. We also sought additional PPI feedback on the intervention content and design from two young people with eczema and a panel of PPI contributors with an interest in skin research, most of whom had experience of eczema, and some were aged 18–25. We discuss the specific contributions of the PPI to intervention development throughout this manuscript. Two young people reviewed the study participant information sheet to check comprehension. One PPI member (AR) discussed and provided feedback on our interpretations of the findings and this manuscript. She continues to help us to disseminate our research findings among her wide-reaching patient networks and via social media.

## PHASE I: INTERVENTION PLANNING
### Phase I: methods

Phase I comprised of three steps: (1) defining the intervention target behaviours; (2) collating and synthesising evidence relating to the perspectives of young people with eczema; and (3) creating an intervention plan.

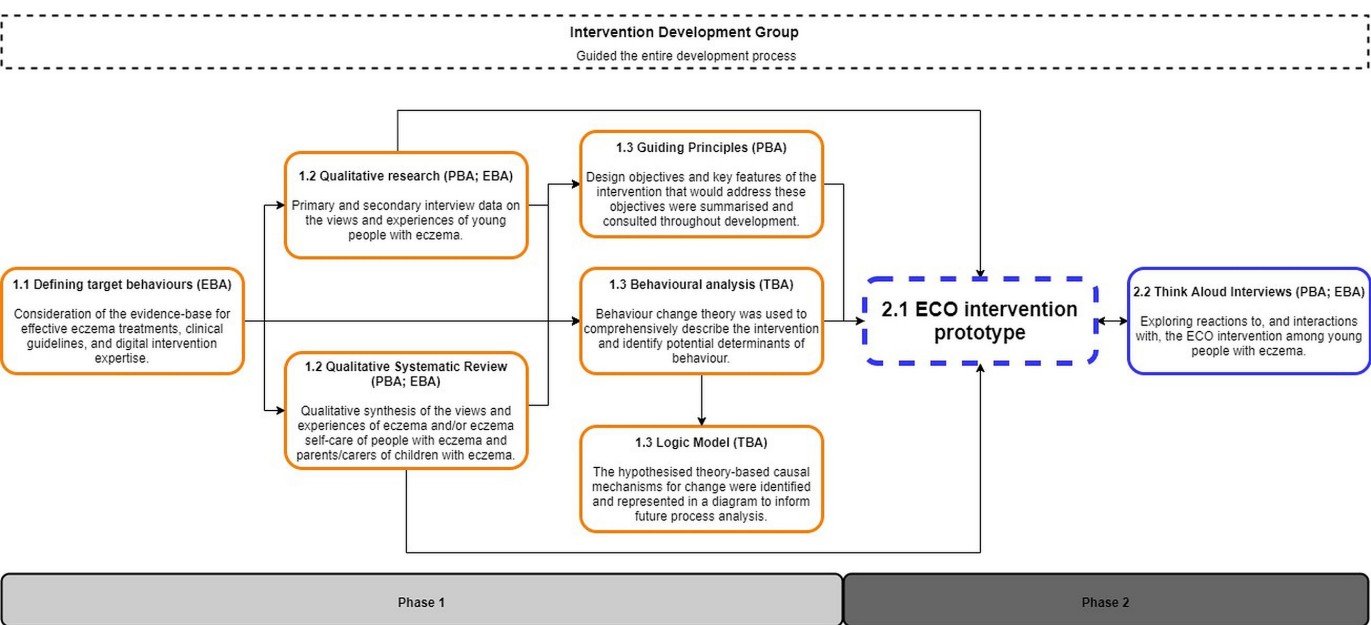

**Figure 1** Intervention development process for Eczema Care Online (ECO) intervention. EBA, evidence-based approach; PBA, person-based approach; TBA, theory-based approach.

### Defining the intervention target behaviours

The multidisciplinary intervention development group agreed the intervention's key target behaviours through consideration of the evidence base for effective eczema treatments, clinical guidance on eczema management and recommendations around what would be feasible and acceptable to implement through an online intervention.[4 23 24]

### Collating and synthesising evidence relating to the perspectives of young people with eczema

First, we undertook a systematic review and thematic synthesis of the views and experiences of eczema and/or eczema self-care of people with eczema and parents or carers of children with eczema.[8] Qualitative studies focusing on the views and experiences of eczema and eczema treatments, and barriers and facilitators to eczema self-management were included. The review identified 39 papers (reporting 32 studies; 9 including young people in the sample).

Second, we carried out a secondary analysis of interview data from 25 young people (17–25 years) with eczema.[5 6 25] The data came from the SKINS project, which explored young people's experiences of living with common skin conditions (eczema, acne, psoriasis, alopecia). To gather additional views from younger adolescents, the current ECO study added to this data set by carrying out interviews with five young people with eczema aged 13–16 years. Both interview studies explored young people's views about eczema treatment and management, and information and support needs. Interviews were analysed using inductive thematic analysis.

The methods and findings of these studies have been described in detail elsewhere.[5 6 8 25] Only key findings relevant to intervention development are summarised in this paper, with the focus of this paper being on intervention development and the novel findings from phase II.

### Developing an intervention plan

Consistent with the person-based approach, our in-depth understanding of young people with eczema informed the development of guiding principles, which outlined key intervention objectives and design features that will address these.[20] A list of potential barriers and facilitators to the target behaviours were also identified from this evidence base and from consultation with the multidisciplinary intervention development group and additional PPI representatives. A behavioural analysis outlined the intervention components that were added to address each of the identified barriers and facilitators for each target behaviour. Consistent with the approach taken by Band et al and Greenwell et al,[17 18] these components were mapped onto behaviour change theoretical frameworks to describe the planned intervention content and identify hypothesised mechanisms of action. The behaviour change techniques taxonomy classifies intervention content by their behaviour change techniques, the smallest component for changing behaviour.[26] The behaviour change wheel was used to classify the source (component of the COM-B model hypothesised to influence behaviour; capability, opportunity, motivation) and function (eg, 'education', 'persuasion') of each individual or group of behaviour change techniques.[27] We also mapped the behaviour change techniques onto their theoretical constructs (eg, 'knowledge', 'skills') using the theoretical domains framework,[28] which is recommended for use alongside the behaviour change wheel.

To illustrate key elements of the intervention's programme theory, a logic model was developed to illustrate how the intervention components, theoretical

constructs (intervention processes) and key behaviours (purported mediators) influence the intervention outcomes (eczema severity).

## Phase I: results
### Defining the intervention target behaviours

The multidisciplinary intervention development group agreed that ECO would aim to reduce eczema severity by supporting young people with eczema to: (1) increase their use of emollients to maintain skin hydration and prevent flare-ups; (2) improve their use of topical corticosteroids or TCIs through reactive applications of these treatments in response to flare-ups or, where appropriate, regular intermittent ('weekend') preventative applications of topical corticosteroids or TCIs if emollients are insufficient as maintenance therapy; (3) improve their management of irritants and triggers; (4) improve their emotional management; and (5) reduce scratching. Use of emollients and topical corticosteroids/TCIs were identified as core behaviours that would likely have the greatest effect on eczema severity. Therefore, intervention content relating to these behaviours was deemed the most important.

### Collating and synthesising evidence relating to the perspectives of young people with eczema

The qualitative evidence helped us to develop the following understanding of our target group. Young people are developing a sense of independent identity and, specifically, young people with eczema are keen to take on more responsibility with their eczema management.[25] However, these young people may find their new roles and responsibilities, such as interacting with health professionals and negotiating healthcare systems, daunting.[25] Young people do not have a comprehensive understanding of eczema, specifically some have little knowledge of the causes/triggers of eczema and the rationale behind their treatment (eg, difference between emollients and topical corticosteroids/TCIs and how to use them).[5 8] Health professionals do not always revisit such information, assuming that young people were told this information in childhood.[5 8] In general, young people perceive topical treatments to be effective, but they also have doubts about their long-term effectiveness, and concerns around their safety and becoming over-reliant on topical corticosteroids.[8 25] This group report several perceived barriers to applying topical treatments, including using treatments when outside of the home (eg, when in class or working in a public-facing job) and cost of treatments.[8 25] These treatment barriers are not unique to this age group,[8] but are nonetheless important to address for any eczema behavioural intervention.

Many young people were told in childhood that they would 'grow out of' eczema, but this information is often at odds with their own experiences.[5] Young people have a desire to 'fit in' with their peers and feel like a 'normal' young person.[6] They welcome the opportunity to share experiences with other young people with eczema to normalise their experience.[6]

The multidisciplinary intervention development group discussed what specific website design needs young people may have. Usability research has shown that this group is likely to dislike reading large amounts of text, such as concepts to be illustrated visually, and relate to content created by peers (eg, stories, images, examples from other young people).[29 30] This evidence was supported by discussions with our young people PPI who also suggested that young people may prefer videos to reading lots of text and recommended having brief 'top tips' with suggestions from other young people for how they manage eczema. They also felt that it was important that the intervention could be accessed via a mobile device or computer.

### Developing an intervention plan

We developed a set of guiding principles to address the issues identified in the evidence synthesis stage (table 1). The behavioural analysis table is presented in online supplemental material 1 and the logic model in figure 2.

## PHASE II: INTERVENTION OPTIMISATION
### Phase II: methods
#### Creating the intervention prototype and videos

Creating the intervention prototype was done in several stages. First, guided by our target behaviours, guiding principles and qualitative research, the multidisciplinary intervention development group agreed the topics of the intervention modules and videos. Second, we wrote page content and video scripts, and circulated this to the multidisciplinary intervention development group for comment to ensure it was evidence based and medically accurate, and the advice was clear and feasible. Positive and negative feedback was entered into the person-based approach table of changes,[16] and potential changes were discussed, agreed and prioritised.

We tested either the video scripts, audio recordings of the scripts or a storyboard or prototype of the video with young people using think-aloud interviews, and these were also reviewed by a PPI panel. Once the written content and videos were finalised, we created a working prototype of the intervention using the LifeGuide software,[31] which was reviewed by our two young people PPI and further optimised through think-aloud interviews with young people with eczema. The final videos were created by an external video creator.

#### Think-aloud interviews

We carried out 30 think-aloud interviews with 28 young people with eczema (2 people took part in 2 interviews, viewing a later intervention version) who were purposively sampled based on age, gender and eczema severity (table 2). Twenty-three of these participants were newly recruited for this study and five participants were from the qualitative study in phase I. Participants were recruited

**Table 1** Guiding principles of Eczema Care Online for young people

| User context | Intervention design objectives | Key intervention features |
|---|---|---|
| ► Young people (YP) with eczema have an increasing desire for autonomy regarding their eczema management, but may feel apprehensive about their new roles and responsibilities.<br>► YP may have gaps in their understanding of eczema.<br>► YP may perceive barriers to using topical treatments. | To support YP to gain autonomy and competence in their eczema management. | ► Ensure YP have a complete understanding of eczema and the rationale behind their treatment.<br>► To build YP's self-efficacy for the target behaviours (eg, information on how to apply treatments, avoid irritants/triggers, reduce scratching).<br>► Stories and tips from other YP on what helped them take control of their own eczema and how to overcome barriers to treatments.<br>► Use autonomy-promoting language, provide choice wherever possible, and avoid condescending or 'child-like' language/graphics.<br>► Provide advice on how to communicate with health professionals and make the most out of appointments. |
| ► YP have a desire to live as 'normal' life as possible.<br>► YPs receive unhelpful messages that eczema is solely a childhood illness | To enable YP to maintain a sense of normalcy when managing their eczema | ► Provide age-appropriate advice on living with eczema (eg, shaving, wearing make-up and managing eczema at work/university/school).<br>► Provide relatable stories and advice from other YP with eczema.<br>► Acknowledge that, for some, eczema persists into adolescence and adulthood.<br>► Provide images and descriptions of eczema for different skin types.<br>► Avoid providing overly restrictive advice on irritants/triggers, instead offering advice on how to minimise the negative consequences of exposure irritants and triggers or provide alternatives (eg, using emollients in place of soap). |
| ► YPs may have doubts and safety concerns about topical treatments.<br>► YP may find topical treatments unpleasant in texture and/or smell and they may worry about applying treatments in public in case others 'found out' that they had eczema. | To build YP's beliefs in the positive effects of their topical treatments | ► Provide information to address topical treatment concerns and barriers, and persuade YPs of the long-term effectiveness of these treatments. |
| ► YPs prefer content that is easy to scan, visual and peer created.<br>► YPs want interventions that are accessible on their mobile devices and computers. | To provide engaging and accessible intervention content | ► Provide interactive content (eg, quizzes), videos and pictures, and reduce reading burden by keeping the amount of text per page to a minimum.<br>► Break the content down into lots of short sections/modules.<br>► Intervention to be mobile friendly.<br>► Provide peer-created content (eg, stories, videos) |

via an invitation letter from their general practice (n=20) or advertising, opportunistic and snowball sampling of students at the University of Southampton (n=8). During the think-aloud interviews, participants were asked to use sections of the intervention while sharing their thoughts and reactions to the content and design aloud. Interviews were facilitated by a researcher who observed the participant using the intervention and asked prompts when needed to elicit participant reactions. At the end of the think-aloud interview, participants were asked some open-ended questions to elicit their general views of the intervention content and design and how it compares to other websites they have used.

To maximise participant time during the interviews, participants testing optional modules were sent the core intervention content to look through before the interview and they were asked about it during the interview. Interviews were carried out at participants' preferred location (eg, at home, the university) and relatives were present for nine interviews. Interviews lasted 45–90 min, took place between October 2018 and April 2019, and were carried out by DG and two medical students (HLAJ, EW) and one postdoctoral student were trained and supervised by DG and ET (postdoctoral experienced qualitative researchers) (all females). Interviews were audio recorded and transcribed verbatim.

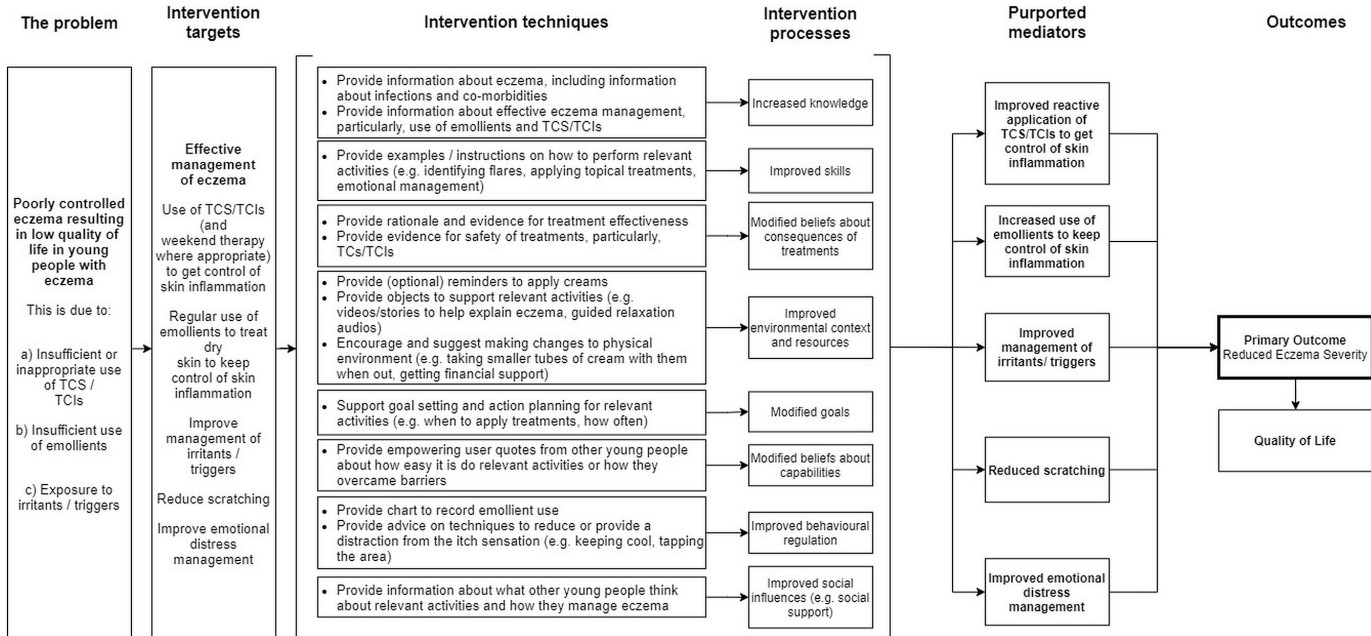

**Figure 2** Logic model for Eczema Care Online (ECO) for young people. **Key:** TCS = Topical Corticosteroids; TCIs = Topical Calcineurin Inhibitors.

Optimisation was iterative, moving between data collection, analysis and intervention modification. We considered data saturation to be reached once no further important changes were required.[16] For each interview, DG completed a feedback table, highlighting positive and negative comments about the intervention, based on the interviewer field notes and listening back to the audio recordings.

MSanter and IM read through the transcripts to ensure important issues were captured. The feedback table was reviewed by a subgroup of the intervention development group (KG, IM, MSanter, KS, MSteele, ET) at weekly meetings. KG transferred negative comments into the table of changes and potential changes were agreed and prioritised by this subgroup.[16]

### Phase II: results
#### Creating the intervention prototype and videos
As the primary focus of the intervention was educational, a website that was also accessible via a mobile device was deemed the most appropriate delivery format. In the final intervention prototype, users first progressed through a brief (nine pages) introductory section containing the key behavioural messages necessary for facilitating a basic understanding of eczema and its management (online supplemental material 2). Some key behavioural messages were also summarised in short (2 min) videos (online supplemental material 3). Users then had the option of completing a simple eczema assessment that provided advice on which of the core treatment modules (emollients or topical corticosteroids) would be most relevant to them, depending on whether they were currently experiencing an eczema flare-up (online supplemental material 4). Users had access to a menu where they could choose the topic modules that were most relevant to them (online supplemental material 5). On each login, users were given suggestions for which modules to look at next, based on what they had looked at previously. Users could choose to have additional behaviour change content delivered by email or SMS text messages.

A key design feature highlighted in the guiding principles was the use of quotes from other young people with eczema sharing their experiences of eczema and eczema management advice (online supplemental material 4). We also introduced new terminology to help young people better understand the function of each type of topical treatment and to mirror how this group already

| Table 2 | Think-aloud participant demographics |
|---|---|
| **Variable** | **Statistic** |
| Age | |
| Mean (SD) | 17.82 (3.41) |
| Range | 13–23 |
| Gender | |
| Female, N (%) | 13 (46.43) |
| Male, N (%) | 15 (53.57) |
| Eczema severity (self-defined)* | |
| Mild, N (%) | 10 (38.46) |
| Mild/Moderate, N (%) | 2 (7.69) |
| Moderate, N (%) | 7 (26.92) |
| Moderate/severe, N (%) | 3 (11.54) |
| Severe, N (%) | 4 (15.38) |

*n=26.

referred to these treatments (eg, 'creams' for all treatments, regardless of whether they were actually a cream or an ointment). Emollients were termed 'moisturising creams' and topical corticosteroids/TCIs were termed 'flare control creams'.

Our multidisciplinary intervention development group and PPI representatives felt that it was important that the intervention is accessible and relevant to all ethnic groups. Therefore, ECO provided images and descriptions of eczema for different skin types and our videos included cartoon characters from different ethnic groups. The full intervention content is outlined in online supplemental material 6 using the TIDieR checklist.[32]

### Think-aloud interviews

Generally, young people expressed positive views of the intervention's content and design. They found the information and advice clear, easy to follow, helpful and relatable, and they liked the videos and brief eczema assessment. Specifically, they found the quotes and advice from other young people with eczema, and the facts about how common it is to still have symptoms at their age reassuring, personal and it made them feel less alone.

> I think it's quite comforting to actually know that it's normal for people with eczema. As in like this is, it's not just me, because even though you get told loads of people have eczema, it is just quite nice to be like, 'This is what you're going through, it's okay, this is what you're going to do about it,' and just have the information all there. (P10, 19–21 years old)

> I like this page (content on prevalence of eczema among young people)…it's good to know that, like, I don't know, I always feel quite weird, because I'm 21 and still have eczema, so, it's good to know that. (P17, 19–21 years old)

Young people valued that ECO provided advice on living well with eczema, rather than focusing solely on medical treatments.

> Sleep problem, emotion, diet and how much you eat - that's good because it's saying that it's not just eczema like using creams, it's also got other stuff, other aspects to it as well…I think you can't just control eczema with creams, you've got to control the whole lot. Yes, this is useful. (P13, 19–21 years old)

> In lighter skin eczema may look red, in darker skin eczema may look grey, purple or brown'. I feel like saying that is actually good, because a lot of the time I'm just like, why does my eczema look grey? But I didn't even know that, now I can see that's a common thing in darker skin. I feel like having that is actually really good. (P22, 16–18 years old)

> I think that (module on itching) was the most helpful because it said apart from using the creams and things, other things you can do to help you. (P19, 13–15 years old)

Most young people explained how they learnt something new about eczema and its management from the part of the website they used, with some explaining that ECO helped addressed the knowledge gaps from childhood.

> (ECO is) brilliant. It's given me a lot more information than I've ever had in the past…I've learnt a lot of new things today about eczema that I didn't know over the last 22 years of having eczema, it's amazing!…It's amazing, I love it. (P9, 22–25 years old)

> If I was prescribed stuff by my doctor I would be like, 'Well, it's probably good for me.'…I'd be like why am I even using this and then not bother. I think if you knew that, okay this one's going to stop you itching and reduce soreness then you might be more likely to carry on using it and also, yes if you're someone that doesn't really ask questions it's nice to have a place that will tell you the information anyway without you having to ask someone. (P18, 16–18 years old)

Young people also talked about how they found the information about the safety of topical corticosteroids reassuring:

> That's good to know, that it (topical corticosteroids) doesn't affect growth or development…because I've been using it for so many years now. It's always been on my mind, and not being able to find out any information about it, it's worried me…but this helps a lot. It kind of puts your mind at ease, knowing that it doesn't do anything to your growth or development. (P9, 22–25 years old)

The positive feedback provided support for our guiding principles, which emphasised the importance of maintaining a sense of normalcy when managing eczema, addressing knowledge gaps to support young people to gain autonomy and competence in eczema management, and providing engaging and accessible intervention content (table 1). The negative feedback was used to modify the intervention (table 3 outlines some example negative comments and the changes implemented).

### DISCUSSION

This manuscript provides a description of the rigorous development process for a behavioural intervention for young people with eczema; a target group that has been largely ignored in eczema research.[8 10 11] It is essential to provide self-management support to this age group as they begin to take on a more active role in their eczema management; a role that was previously taken on by their families.[7] The person-based approach to intervention development allowed us to understand and accommodate the perspectives of young people with eczema, resulting in an online intervention that was engaging and acceptable to our sample of this target group.[20]

**Table 3** Example issues identified from the think-aloud interviews and the changes implemented to address these

| Summary of issue identified | Example quote | Change implemented |
|---|---|---|
| Some found the questions in the brief eczema assessment confusing. | 'Does your skin feel dry?' Well, everywhere or in general where you've got…eczema? …I don't know if that's (the question) very clear. (P12, 22–25 year old)<br>'Is your skin or redder or darker than usual?'…to me it sounds like 'is your eczema darker or less dark than your eczema normally is?', but I think it's a bit ambiguous whether it means that or their usual skin?….you could just say 'is your eczema red or darker than usual?' (P18, 16–18 years old)<br>'Is your skin redder or darker (than usual)?'… maybe it's the 'than usual' bit that needs to be more clearer, because I could interpret that as this being my usual skin, rather than the usual colour of the eczema…maybe, 'Is your eczema redder or darker than usual?' or something (P5, 19–21 years old) | The assessment questions were reworded to clarify that we are asking about their eczema at present (rather than in the past) and their eczema skin (rather than skin in general). |
| The feedback for the brief eczema assessment did not always match their experience (eg, feedback suggested they may have an eczema flare-up when they did not). | (Is your skin itchy or sore?) it's not really sore but it is itchy…I can have itchy dry skin but I only have sore skin when it's red and inflamed… (recommended the flare control creams module)… it said that I'm having a flare-up now, but I wouldn't class this as a flare-up. This is kind of just my - how my eczema kind of bobbles along, as in my flare-ups would be much more aggressive than what they were suggesting it is. (P10, 19–21 years old) | We separately asked whether eczema is 'itchy' and 'sore' so that people do not receive feedback that they are having a flare-up if they just have itchy and not sore skin. Softened the feedback on current eczema severity to avoid disengaging those who do not agree that they are experiencing an eczema flare-up ('for most people, this means they are having an eczema flare-up'). |
| Some thought the introductory section was too long. | Personally, if I see there's 21 pages, I'm just going to try and get through them. (P18, 16–18 years old) | Reduce the introductory section significantly with optional click-outs to additional information if needed. |
| Some young people commented that they already knew a lot of the information as they have had eczema for a long time. | (ECO) was really good…because I've had it for so long I feel like a lot of the information there I've already had drilled into me my whole life, but I feel like for people that either have children with eczema or people with eczema, I don't know, I think it would be really helpful… personally, I would probably want the more information about the lifestyle stuff and diet (P10, 19–21 years old)<br>I think a lot of stuff I probably kind of already knew from having eczema but it's good to get a few things that help yes (P19, 13–15 years old) | Added in a quote from another young person who had eczema for a long time saying how they were surprised that the website contained new helpful tips. Emphasise that the information is based on the most up-to-date research evidence (so they may find new information) and that there is information on 'living with eczema' (eg, diet advice). |

The behavioural analysis maintained a focus on the behaviours most likely to influence eczema severity (topical treatment use, managing irritants/triggers, scratching, emotional management) and suggested acceptable behaviour change techniques that can help young people with eczema overcome behavioural barriers relevant to them. Logic modelling offered a programme theory that can be tested and refined in future process analyses. Our final guiding principles outlined some of the key behavioural issues, intervention design objectives and design features that those developing behavioural interventions for young people with eczema, or other long-term conditions, may wish to consider to maximise user engagement. The think-aloud interviews provided support for the relevance of these guiding principles to this target group and the acceptability of the design features we implemented to engage this group. Specifically, the ECO intervention's use of quotes and tips from other young people with eczema and its focus on living with eczema (not just topical treatments) were valuable for normalising eczema.

The value of behaviour change interventions that make individuals feel they are 'not alone' is supported by other qualitative research with people with long-term conditions and parents/carers of children with eczema.[23 33] Although this intervention focused mainly on treatment adherence as this was deemed to have the biggest influence on eczema severity, providing age-appropriate

advice and support on how young people can better live with eczema was valuable for ensuring the content was engaging and relatable to this target group. Young people explained how valuable the explanations of eczema and rationale behind the topical treatments for addressing gaps in their knowledge, with most participants reporting that they learnt something new from ECO. This provided further support for the need for health services to revisit eczema education with young people, avoiding assumptions that they have already been told this information.[5] Online information can be confusing and of variable quality,[34 35] therefore, it is important to signpost young people to high-quality evidence-based online information so they are empowered to take an active role in their own healthcare.[36]

A strength of this research was our rigorous development using complementary theory-based, evidence-based and person-based approaches to ensure the intervention is acceptable and engaging to its target group. Our multidisciplinary intervention development group and PPI ensured that the content was evidence based, that advice was feasible and that the perspectives of people living with eczema were considered throughout the whole development process.[37] It was helpful to gain iterative feedback from PPI and young people with eczema on early versions of the videos (ie, scripts, audio recordings of the video voiceover, prototype) to ensure that they were as acceptable as possible to the target group before it was finalised by the external video creator. One limitation of the think-aloud interview study is that we did not collect information on participant ethnicity, socioeconomic status or educational level. It will be important to purposively sample based on these demographics in future evaluations of this intervention to ensure it is engaging and effective for all participant groups and ensure digital interventions do not further facilitate healthcare inequalities. Furthermore, as think-aloud interviews explore participants' immediate reactions to the intervention, they cannot tell us how people would use the intervention over time or explore engagement with the target behaviours.

In conclusion, the think-aloud interview study demonstrated ECO is likely to be acceptable to young people with eczema and provided support for our guiding principles, including key design objectives and features to consider when developing interventions for this population. A randomised controlled trial of ECO to explore effectiveness and cost-effectiveness is underway (ISRCTN79282252).[14] This includes a nested mixed methods process evaluation to explore young people's experiences of using the intervention over time and how the intervention was used, and test and further refine the intervention's programme theory. For the qualitative interview study, we purposively recruited participants across a range of ages, ethnicities, socioeconomic status, eczema severities and intervention usage.

**Author affiliations**
[1]Centre for Clinical and Community Applications of Health Psychology, Faculty of Environmental and Life Sciences, University of Southampton, Southampton, UK
[2]Division of Psychology & Mental Health, School of Health Sciences, Faculty of Biology, Medicine and Health, The University of Manchester, Manchester, UK
[3]Department of Psychology, Faculty of Science and Health, University of Portsmouth, Portsmouth, UK
[4]School of Primary Care, Population Health and Medical Education, Faculty of Medicine, University of Southampton, Southampton, UK
[5]Population Health Sciences, University of Bristol, Bristol, UK
[6]Centre of Evidence Based Dermatology, School of Medicine, University of Nottingham, Nottingham, UK
[7]Department of Dermatology, Rotherham NHS Foundation Trust, Rotherham, UK
[8]Faculty of Epidemiology and Population Health, London School of Hygiene & Tropical Medicine, London, UK
[9]Faculty of Health Education and Life Sciences, Birmingham City University, Birmingham, UK
[10]School of Life and Medical Sciences, University of Hertfordshire, Hatfield, UK
[11]Faculty of Medicine, University of Southampton, Southampton, UK
[12]Centre for Academic Primary Care, School of Psychological Science, University of Bristol, Bristol, UK

**Acknowledgements** We would like to thank our young people PPI Taheeya and Tahmid Ahmed and the panel of public contributors for their valuable feedback on the Eczema Care Online intervention.

**Contributors** MSanter is the guarantor. KG, DG, KS, MSteele, ET, MJR, AR, JRC, SLawton, SLangan, FC, SW, HCW, KST, LY, MSanter and IM designed the study. KG, KS, DG, MSteele, IM and MSanter led the intervention development with input from ET, MJR, AR, JRC, SLawton, SLangan, FC, ELR, SW, HLAJ, EW, HCW, KST, LY, MSanter and IM. DG, ET, SW, HLAJ and EW were responsible for recruitment and data collection. KG and DG led on the data analysis, with support from MSanter and IM. KG drafted the manuscript, with initial support from DG, MSanter and IM. All authors critically reviewed the manuscript, contributing important intellectual content and approved the final manuscript.

**Funding** This article presents independent research funded by the National Institute for Health Research (NIHR) under its Programme Grants for Applied Research (PGfAR) Programme (Grant Reference Number RP-PG-0216-20007). Eczema Care Online (ECO) was developed using the LifeGuide software, which was partly funded by the NIHR Biomedical Research Centre (BRC), Southampton. The views expressed are those of the authors and not necessarily those of the NIHR or the Department of Health and Social Care. SLangan is funded by a Wellcome Senior Clinical Research fellowship (205039/Z/16/Z). This research was funded in whole or in part by the Wellcome Trust (205039/Z/16/Z). For the purpose of Open Access, the author has applied a CC BY public copyright licence to any Author Accepted Manuscript (AAM) version arising from this submission.

**Competing interests** SLangan is a coinvestigator on the IMI Horizon 2020 project BIOMAP, but is not in receipt of industry funding.

**Patient and public involvement** Patients and/or the public were involved in the design, or conduct, or reporting, or dissemination plans of this research. Refer to the Intervention development methodology section for further details.

**Patient consent for publication** Not applicable.

**Ethics approval** This study involves human participants and was approved by Wales REC 7 Ethics Committee (REC 17/WA/0329) Participants gave informed consent to participate in the study before taking part.

**Provenance and peer review** Not commissioned; externally peer reviewed.

**Data availability statement** Data are available upon reasonable request. The data that support the findings of this study are available from the corresponding author upon reasonable request.

for any error and/or omissions arising from translation and adaptation or otherwise.

**ORCID iDs**
Kate Greenwell http://orcid.org/0000-0002-3662-1488
Daniela Ghio http://orcid.org/0000-0002-0580-0205
Katy Sivyer http://orcid.org/0000-0003-4349-0102
Emma Teasdale http://orcid.org/0000-0001-9147-193X
Matthew J Ridd http://orcid.org/0000-0002-7954-8823
Joanne R Chalmers http://orcid.org/0000-0002-2281-7367
Sinead Langan http://orcid.org/0000-0002-7022-7441
Fiona Cowdell http://orcid.org/0000-0002-9355-8059
Kim Suzanne Thomas http://orcid.org/0000-0001-7785-7465
Ingrid Muller http://orcid.org/0000-0001-9341-6133

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
