## [Reviewer comments · BMJ Open]

ARTICLE DETAILS

TITLE (PROVISIONAL)	Eczema Care Online: development and qualitative optimisation of an online behavioural intervention to support self-management in young people with eczema
AUTHORS	Greenwell, Kate; Ghio, Daniela; Sivyer, Katy; Steele, Mary; Teasdale, Emma; Ridd, Matthew; Roberts, Amanda; Chalmers, Joanne; Lawton, Sandra; Langan, Sinead; Cowdell, Fiona; Le Roux, Emma; Wilczynska, Sylvia; Jones, Hannah; Whittaker, Emilia; Williams, HC; Thomas, Kim; Yardley, Lucy; Santer, Miriam; Muller, Ingrid

VERSION 1 – REVIEW

REVIEWER	Tso, Simon Warwick Hospital, Jephson Dermatology Centre
REVIEW RETURNED	14-Sep-2021

GENERAL COMMENTS	I read this manuscript with interests. It is a thoughtful and well written manuscript describing the development of the ECO resource for younger persons, and how the resource was optimised based on feedback. The strength of this manuscript is its detailed description of its methodology (down to the finer detail such as secondary analysis of data from SKINS project) and the extend of patient and public involvement in the development, interpretation and dissemination of the work. The work described in this manuscript is highly relevant to patients and clinicians. It is very well written and this is one of the very few situations where I have no further suggestions to recommend to the authors.
---

REVIEWER	Diwakar, Lavanya University of Birmingham, Health Economics Unit
REVIEW RETURNED	04-Oct-2021

GENERAL COMMENTS	This is a very interesting paper, describing a potentially very useful educational platform for young people with eczema. There is strong PPI involvement, which improves the relevance of patient facing platforms like this. The paper is well written and is quite detailed. I have a few minor comments. 1. The descriptions are very long and could do with a little rephrasing. Some of the descriptions are repeated a few times- for example, the intervention development methodology is described in some detail in the introduction section and the methods section; the results are discussed in detail in sections 1.1-1.3 and again in a table.
---

	2.Consider deleting description of methods from introduction section 2.Discussion: Page 24, line 29: Suggest omit the phrase "as well as effective"- since effectiveness was not evaluated in this study.
--	---

REVIEWER	Chovatiya, Raj Northwestern University, Department of Dermatology
-----------------	--

REVIEW RETURNED	28-Nov-2021
-------------

GENERAL COMMENTS	Overall a very well written study about an important topic that is under-explored in atopic dermatitis - how to truly implement care strategies that allow us to educate and change behavior beyond the couple of minutes the patient may have with their healthcare provider. A few points: Line 6 – I would explain that eczema is being used to describe the most common type of eczema, atopic dermatitis (or atopic eczema), for the purposes of this study – these are what the citations are all referring to as well. It's understandable the terminology 'eczema' is being emphasized given the patient centered nature of the research and ECO tool. Lines 8-16 – basic treatment of all eczema includes optimization of bathing (not mentioned here) alongside optimized moisturization and trigger avoidance – would be important to mention this third part of this basic strategy as has been described by the AAD, ETAF, and nicely summarized in the AD Yardstick. For the purposes of the entire paper and the ECO tool, I think that bathing would be an important topic to emphasize through education and patient motivation. It's a very basic strategy and we have a good body of literature on this topic. It has not been given much attention in your current draft. Additionally for the purposes of this entire paper and tool, topical anti-inflammatories are a treatment strategy for mild eczema along with moderate and severe disease – not just starting at moderate (as stated in the introduction, even if this is referring to TCI, which is also not entirely accurate). This includes topical corticosteroids and topical calcineurin inhibitors along with topical PDE4 inhibitor and most newly, topical JAK inhibitors. Topical treatments are used both proactively AND to treat flare ups, therefore they are used to keep control and get control, respectively - and this confuses me slightly based on how this is discussed with the text, supplemental figures, and ECO tool as a delineation between moisturizers and topical anti-inflammatories.
---

REVIEWER	van Halewijn, Karlijn Erasmus MC, Department of General Practice
-----------------	---

REVIEW RETURNED	26-Jan-2022
-------------

GENERAL COMMENTS	Thank you for the invitation to review this paper about the development of an online behaviour change intervention for young people with eczema and the optimization of the intervention throughout qualitative think-aloud interviews. This paper covers an important topic of great clinical interest. The study is well thought and it is interesting to read all the steps the team carefully made to create and improve the behaviour change intervention. I have some minor comments.
--

	1. Introduction: There is stated that the aim of the ECO program is to develop two online behavioural interventions. How did you decided to choose a website as format for this online intervention? Is a website the most appropriate medium for this target group and the intervention? 2. Intervention development method: did you consider to involve a communication specialist in the multidisciplinary intervention development group? 3. Patient and public involvement (page 6). Are the two mothers as stated here the only participants in the PPI? Are the mothers most representative for the target group of people aged 13-25? Furthermore there is mentioned that additional PPI feedback is sought by two young people with eczema. Line 44-47 is stated that AR discussed an provided feedback, is it possible to explain her role in more detail? Is she a patient? Or is she a researcher? 4. Phase 2. Methods. Who was involved in the design of the website? In addition to the content, which has been collected in an extensive manner, the design is also very important for this target group. 5. Phase 2. Results. Page 16 line 37. It is mentioned that that the intervention should be accessible and relevant to all ethnic groups. Is ethnic diversity covered throughout the participants of the qualitative think-aloud interviews and taken into account when sampling? If this is not the case, maybe add this to the discussion section. Furthermore it is mentioned that the education level of the participants of the qualitative think-aloud interviews is a limitation. How many of the participants where high educated students? Can you please clarify education level in table 2. Now it is only stated in the discussion and not elsewhere. 6. 2.2 Think-aloud interviews. It seems that the method of sampling has become more of convenience sampling because, as mentioned below in the paper, many highly educated students took part. See also comment 5. This could be a serious problem for the generalizability. 7. 2.2 Think-aloud interviews. Were the participants asked whether the website met their desire? And whether the design matched what they wanted? Would the participants visit the website in 'real life'? 8. General. Is there already a plan for implementation and/or dissemination? If it appears from the RCT that the intervention is not effective, you have put a lot of time and effort into this website. It would be really sad if you did not spread this science-based content. So how would you integrate it into the healthcare system? Very curious about the website though and hopefully the trial will reveal some good results! Acknowledgement: completed the review with help from Gijs Elshout.
--	--

VERSION 1 – AUTHOR RESPONSE

Reviewer 1 comments:

1. I read this manuscript with interests. It is a thoughtful and well written manuscript describing the development of the ECO resource for younger persons, and how the resource was optimised based on feedback. The strength of this manuscript is its detailed description of its methodology (down to the finer detail such as secondary analysis of data from SKINS project) and the extend of patient and public involvement in the development, interpretation and dissemination of the work. The work described in this manuscript is highly relevant to patients and clinicians. It is very well written and this is one of the very few situations where I have no further suggestions to recommend to the authors.

RESPONSE: We thank you for reviewing our manuscript and for your positive comments.

Reviewer 2 comments:

2. The descriptions are very long and could do with a little rephrasing. Some of the descriptions are repeated a few times- for example, the intervention development methodology is described in some detail in the introduction section and the methods section; the results are discussed in detail in sections 1.1-1.3 and again in a table.

3. Consider deleting description of methods from introduction section

RESPONSE: Thank you for pointing out this repetition. To address the above two comments, we have now removed all mentions of intervention development methodology from the introduction, and this now reads: "First, we aimed to describe the development of the ECO intervention for young people with eczema (Phase 1). Second, we aimed to explore and optimise the acceptability of the ECO intervention among young people with eczema (Phase 2). This article highlights key psychosocial needs of young people with eczema and intervention features to consider when developing behavioural interventions for this group."

We have rephrased or removed repetitive content from 1.2 and Table 1. We have decided to keep the results about user context in Table 1 (column 1), as this is an important column in the Person-Based Approach guiding principles table that demonstrates how the intervention objectives and design features map onto the evidence-base about user context. However, we have rephrased some of the statements in this column, so it is simply a summary of what was reported in detail in 1.2.

4. Discussion: Page 24, line 29: Suggest omit the phrase "as well as effective"- since effectiveness was not evaluated in this study.

RESPONSE: Suggested change made.

Reviewer 3 comments:

5. Line 6 – I would explain that eczema is being used to describe the most common type of eczema, atopic dermatitis (or atopic eczema), for the purposes of this study – these are what the citations are all referring to as well. It's understandable the terminology 'eczema' is being emphasized given the patient centered nature of the research and ECO tool.

RESPONSE: The reviewer makes a good point, and we appreciate that in some countries, eczema can refer to a group of conditions characterised by skin inflammation. However, this is not the case according to the World Allergy Organisation nomenclature committee, who came up with a very clear and logical classification of diverse form of dermatitis as shown in the figure uploaded with this response.

We recognise that the term atopic dermatitis is widely used in the US and Japan and that it is important for an international readership to make it clear that when we refer to eczema, we are referring to atopic eczema/atopic dermatitis. We have therefore added the following sentence to the start of the introduction to make it clear to readers what condition we are referring to throughout by the use of the term “eczema” and referenced the Report of the Nomenclature Review Committee of the World Allergy Organization to support this statement: “Atopic eczema (syn. atopic dermatitis) is the most common type of dermatitis/eczematous inflammation and will be referred to from here on as just “eczema” in accordance with the nomenclature of the World Allergy Organisation.[1]” Johansson SG., Bieber T, Dahl R, et al. Revised nomenclature for allergy for global use: Report of the Nomenclature Review Committee of the World Allergy Organization, October 2003. *J Allergy Clin Immunol* 2004;113:832–6. doi:10.1016/j.jaci.2003.12.591

6. Lines 8-16 – basic treatment of all eczema includes optimization of bathing (not mentioned here) alongside optimized moisturization and trigger avoidance – would be important to mention this third part of this basic strategy as has been described by the AAD, ETAF, and nicely summarized in the AD Yardstick.

For the purposes of the entire paper and the ECO tool, I think that bathing would be an important topic to emphasize through education and patient motivation. It's a very basic strategy and we have a good body of literature on this topic. It has not been given much attention in your current draft.

RESPONSE: The ECO tool includes advice on bathing, but as the focus of this paper is on the methodology of how the intervention was developed, we have kept description of content brief. However, advice on bathing is referred to as part of the target behaviour "management of irritants and triggers". Table 1 clarifies this as it includes the following text: "advice on how to minimise the negative consequences of exposure irritants and triggers or provide alternatives (e.g. using emollients in place of soap)." Inclusion of content on bathing is also presented in the overview of the key components (Table 1) in Supplementary Material 6.

7. Additionally for the purposes of this entire paper and tool, topical anti-inflammatories are a treatment strategy for mild eczema along with moderate and severe disease – not just starting at moderate (as stated in the introduction, even if this is referring to TCI, which is also not entirely accurate). This includes topical corticosteroids and topical calcineurin inhibitors along with topical PDE4 inhibitor and most newly, topical JAK inhibitors.

RESPONSE: Thank you for this pointing out. We did not mean to infer that topical anti-inflammatories do not have a role in mild eczema so have removed reference to this treatment being ‘for those with moderate to severe eczema’ in the following sentence: “Eczema management focuses on identification and avoidance of irritants/triggers that may exacerbate eczema symptoms; regular use of emollients to restore skin barrier function; and topical corticosteroids or topical calcineurin inhibitors (TCIs) to treat flare-ups.[4]”

As the ECO tool is developed primarily for the UK population, we have based advice on NICE and SIGN guidelines, which do not include topical PDE4 inhibitors or newer topical JAK inhibitors.

8. Topical treatments are used both proactively AND to treat flare ups, therefore they are used to keep control and get control, respectively - and this confuses me slightly based on how this is discussed with the text, supplemental figures, and ECO tool as a delineation between moisturizers and topical anti-inflammatories.

RESPONSE: We agree with the reviewer that in addition to reactive use, topical corticosteroids and topical calcineurin inhibitors can be used proactively (for two consecutive days each week) in order to maintain good atopic eczema control. For the vast majority of milder cases in the community where

flares are less frequent, a reactive approach with topical anti-inflammatory treatments is used, along with emphasis of continuing with emollients to prevent flares. The ECO tools do explain that, for some people, emollients are insufficient for maintenance, in which case proactive intermittent (eg weekend or two consecutive days each week) topical anti-inflammatories may be needed. However, it is more likely that the subgroup with more brittle disease who require topical anti-inflammatories for maintenance are more likely to be under regular dermatological review and receiving supplementary advice to the ECO tools through that route.

Reviewer 4 comments:

9. Introduction: There is stated that the aim of the ECO program is to develop two online behavioural interventions. How did you decided to choose a website as format for this online intervention? Is a website the most appropriate medium for this target group and the intervention?

RESPONSE: We have added in the following into section 2.1 to explain our decision to use a website delivery format:

“As the primary focus of the intervention was educational, a website that was also accessible via a mobile device was deemed the most appropriate delivery format.”

Furthermore, as mentioned on page 11, our young people PPI also felt that it was important that the intervention could be accessed via both a mobile device and computer. We also added in the following showing how we improved accessibility among this target group: “Users could choose to have additional behaviour change content delivered by email or SMS text messages.”

10. Intervention development method: did you consider to involve a communication specialist in the multidisciplinary intervention development group?

RESPONSE: Many of our intervention development group members had extensive experience in writing patient-friendly health information. Therefore, we added in the following to clarify this: “Both phases were guided by a multidisciplinary intervention development group, which comprised 18 members including ... experts in intervention development, writing patient-friendly health information, and long-term conditions in adolescents.”

11. Patient and public involvement (page 6). Are the two mothers as stated here the only participants in the PPI? Are the mothers most representative for the target group of people aged 13-25? Furthermore there is mentioned that additional PPI feedback is sought by two young people with eczema. Line 44-47 is stated that AR discussed an provided feedback, is it possible to explain her role in more detail? Is she a patient? Or is she a researcher?

RESPONSE: The two PPI who were mothers of children with eczema were active members of the intervention development group, which covered the two online interventions (young person and parents/carers of children with eczema). We did not invite any young people to this group as we felt that group meetings could be quite intimidating for this target group. Instead, we felt it was more appropriate to seek feedback from young people PPI on the intervention content outside of these meetings, which we detail in the manuscript. We have added the following to clarify that the intervention development group covered both interventions: “Through regular meetings and reviewing documents, this group...provided detailed feedback on the intervention plans, written content, and prototypes for both online interventions (young people and parents and carers of children with eczema).”

We have clarified the role of AR as follows:

“Two mothers of children and young people with eczema (one of whom had eczema herself and helps run an eczema support group [AR]) ...”.

“One PPI member (AR) discussed and provided feedback on our interpretations of the findings and this manuscript.”

12. Phase 2. Methods. Who was involved in the design of the website? In addition to the content, which has been collected in an extensive manner, the design is also very important for this target group.

RESPONSE: We have added the following to clarify that the multidisciplinary intervention development group, PPI, and think aloud participants were also involved in the website design:

“...this group...provided detailed feedback on the intervention plans, written content, website design, and prototypes for both online interventions”

“We also sought additional PPI feedback on the intervention content and design from two young people with eczema and a panel of PPI contributors”

“During the think-aloud interviews, participants were asked to use sections of the intervention while sharing their thoughts and reactions to the content and design aloud.”

13. Phase 2. Results. Page 16 line 37. It is mentioned that that the intervention should be accessible and relevant to all ethnic groups. Is ethnic diversity covered throughout the participants of the qualitative think-aloud interviews and taken into account when sampling? If this is not the case, maybe add this to the discussion section. Furthermore it is mentioned that the education level of the participants of the qualitative think-aloud interviews is a limitation. How many of the participants were high educated students? Can you please clarify education level in table 2. Now it is only stated in the discussion and not elsewhere.

RESPONSE: Unfortunately, we do not have complete data on ethnic group or educational level for all participants, so we have removed the sentence discussing the educational level of the sample and have discussed the lack of demographic data as a research limitation: “One limitation of the think-aloud interview study is that we did not collect information on participant ethnicity, socioeconomic status, or educational level. It will be important to purposively sample based on these demographics in future evaluations of this intervention to ensure it is engaging and effective for all participant groups and ensure digital interventions do not further facilitate healthcare inequalities.”

14. 2.2 Think-aloud interviews. It seems that the method of sampling has become more of convenience sampling because, as mentioned below in the paper, many highly educated students took part. See also comment 5. This could be a serious problem for the generalizability.

RESPONSE: Only 8 of the 28 participants were recruited via opportunistic and snowball sampling from the University. The remaining 20 participants were recruited via their GP practice. Across both recruitment methods, we employed purposive sampling to ensure we included participants with a range of ages, gender, and eczema severity.

15. 2.2 Think-aloud interviews. Were the participants asked whether the website met their desire? And whether the design matched what they wanted? Would the participants visit the website in 'real life'?

RESPONSE: We have added in the following to clarify that we asked several general open-ended evaluative questions at the end of the interview. “At the end of the think-aloud interview, participants were asked some open-ended questions to elicit their general views of the intervention content and design and how it compares to other websites they have used.”

16. General. Is there already a plan for implementation and/or dissemination? If it appears from the RCT that the intervention is not effective, you have put a lot of time and effort into this website. It would be really sad if you did not spread this science-based content. So how would you integrate it into the healthcare system? Very curious about the website though and hopefully the trial will reveal some good results!

RESPONSE: We thank you for your interest in our research. As part of our research programme, we are currently implementing the two ECO interventions across primary and secondary care and the community (e.g. pharmacies, eczema charities). We are still awaiting the RCT results, but currently feel we have sufficient research evidence from this study and our subsequent mixed methods process evaluation to suggest that the interventions are useful, engaging and acceptable to both target groups.

VERSION 2 – REVIEW

REVIEWER	Chovatiya, Raj Northwestern University, Department of Dermatology
REVIEW RETURNED	25-Mar-2022
GENERAL COMMENTS	Authors have appropriately addressed the reviewer comments.